organic chemistry/green chemistry

eco-conscious reaction, methoximation of aldehydes and ketones, MnCl$_2$-catalysed transformation

**Authors for correspondence:**
Iván Cortés
e-mail: cortes@iquir-conicet.gov.ar
Andrea B. J. Bracca
e-mail: bracca@iquir-conicet.gov.ar

This article has been edited by the Royal Society of Chemistry, including the commissioning, peer review process and editorial aspects up to the point of acceptance.

# Eco-friendly methoximation of aromatic aldehydes and ketones using MnCl$_2$.4H$_2$O as an easily accessible and efficient catalyst

Melina C. Ontivero[1,2], Teodoro S. Kaufman[1,2], Iván Cortés[1,2] and Andrea B. J. Bracca[1,2]

[1]Facultad de Ciencias Bioquímicas y Farmacéuticas, Universidad Nacional de Rosario, Suipacha 531, S2002LRK Rosario, Santa Fe, Argentina
[2]Instituto de Química Rosario (IQUIR, CONICET-UNR), Suipacha 531, S2002LRK Rosario, Santa Fe, Argentina

MCO, 0000-0001-7878-2565; TSK, 0000-0003-3173-2178;
IC, 0000-0002-8069-2793; ABJB, 0000-0003-2647-475X

Methoximes are important as a class of intermediates and products, among fine chemicals and specialties. The development of a new, facile and efficient method for their synthesis is reported. The methoximes were properly accessed from the corresponding aromatic aldehydes and ketones in good to excellent yields, under mild conditions, employing the inexpensive and environmentally friendly MnCl$_2$.4H$_2$O as a catalyst (at low loading and without the addition of ligand), in EtOH at 50°C. The scope of the process was systematically assessed.

## 1. Introduction

Oximation is a very efficient strategy for the derivatization of aldehydes and ketones. The synthesis of methoximes is a highly relevant transformation with both theoretical and practical implications in the chemistry of carbonyl compounds [1]. Methoximation has been used for characterization and purification of carbonyl compounds, for the protection or activation of carbonyl groups in multistep synthesis [2–5]; the reaction has also found use in Heck-type and C–H bond activation reactions, as well as in O–H and N–O bond cleavage, followed by coupling or C–N bond formation [6].

Methoximes are also useful synthetic intermediates and many relevant compounds display this characteristic as a distinguished motif. Figure 1 shows methoxime derivatives, such as **1** [7], the

**Figure 1.** Selection of relevant bioactive methoxime derivatives.

pesticide **2** [8] the antibacterial **3** [9], the fungicidal **4** [10] and the antiproliferative agent **5** [11] are found in the patent literature. They have also been prepared as prodrugs (**6**, anticonvulsant) [12], enzyme inhibitors (**7**, **8**) [13,14], antiparasitic agents (**9**) [15] and receptor agonists (**10**) [16]. Several methoxime derivatives are on the market, such as moxidectin [17].

A variety of carbonyl substrates, including aliphatic, aromatic and heterocyclic, have hitherto been subjected to methoximation. Though most of the existing strategies are efficient in their own way, they exhibit a series of drawbacks, including the usual need of reflux conditions and/or unnecessarily excessive amounts of MeONH₂.HCl (up to 5 equiv.) and base to achieve reaction completion in a short time. They also resort frequently to the use of less than optimal solvents, such as pyridine, and often require rather extended reaction times.

With the advent of biorthogonal chemistry and the increasing interest in the preparation of bioconjugates, during the last few years, several types of organocatalysts for the alkoxime ligation have been studied; these comprise mainly aniline derivatives which operate under different conditions (aqueous media, dilute solutions), and some of them are comparatively expensive [18].

On the other hand, in a recent review, it was pointed out that the metal-mediated catalytic oximation of carbonyl derivatives has been scarcely reported. The authors correctly observed that the reaction was 'still-uncommon' and deceptively, indicated that 'the synthetic potential of many of these approaches is

**Scheme 1.** Proposed MnCl$_2$.4H$_2$O-catalysed synthesis of aromatic methoximes.

not high' [19]. Among the few available alternatives, a titanium complex has been shown to promote the alkoxymation of carbonyl compounds [20], and iron salts have been employed to catalyse the formation of α- and β-hydroxy oximes [21].

Not all the available alternatives suffer simultaneously from all the drawbacks; however, in view of the multiple disadvantages of the available methodologies, we have considered that the development of simple, straightforward, eco-conscious and practical catalytic systems for the transformation of aromatic carbonyls such as ketones and aldehydes into their corresponding methoximes would be highly desirable, because it is a real current need.

Thus, in contrast with the previous conclusion [19], we have recently advanced the state of the art by demonstrating that the methoximation of aldehydes and ketones can be accelerated by various Lewis acids, highlighting CeCl$_3$.7H$_2$O as a suitable promoter, being efficient under mild conditions [22]. During this study, we noted that manganese salts [Mn(OAc)$_3$·2H$_2$O, MnSO$_4$·4H$_2$O, MnCl$_2$·4H$_2$O] were also able to conveniently assist the methoximation of a model ketone, with the chloride salt exhibiting the best performance [22].

Manganese is a widely found transition metal, being the third most abundant among the latter. It is environmentally benign, and its derivatives are particularly attractive, because of their comparatively low cost and low toxicity [23,24]. Therefore, its use in chemical transformations in place of other transition metals contributes to process eco-consciousness and sustainability.

As part of our ongoing research programme, we are interested in developing simple, efficient and eco-conscious transformations and synthetic procedures [25–27]. Therefore, in pursuit of our longstanding goals and in order to shed more light on our previous observations on the promising properties of manganese salts, herein we disclose our results regarding MnCl$_2$.4H$_2$O as a convenient and eco-friendly methoximation catalyst (scheme 1) with a set of 22 examples of known and unknown methoximes, and compare the results with the performance of analogous methoximations at the state of the art, including those which employ CeCl$_3$.7H$_2$O as promoter [22]. The proposed use of MnCl$_2$.4H$_2$O for methoximations is novel and original, since despite its ready availability the salt has been employed rather scarcely in organic chemistry, and mostly in C–C bond-forming processes. In addition, we report on the scope of the reaction and propose a plausible mechanism.

# 2. Material and methods

## 2.1. General information

The reactions were performed in dry glassware, using recently distilled anhydrous solvents. Anhydrous NaOAc was prepared as previously reported [22]; the remaining chemicals were also of analytical reagent (AR) grade and were employed as received.

Supervision of the reactions advance was performed by TLC, where detection of the chromatographic spots was made by irradiation with short wavelength UV light (254 nm). The spots were revealed by careful heating of the plates, after their exposure to the ethanolic p-anisaldehyde/sulfuric acid reagent. Single spots were observed on the TLC plates, developed with hexanes or with various hexanes-EtOAc solvent systems.

The flash chromatographies were run in columns, slurry-packed with a suspension of silica gel 60 H (particle size 63–200 µm) in hexanes. The elution was performed under a positive pressure of air (0.9–1.1 atm.), applying gradient of solvent polarity (hexane-EtOAc mixtures) procedures.

## 2.2. Apparatus

The melting points, GC-MS, FT-IR, as well as one-dimensional and two-dimensional nuclear magnetic resonance (NMR) spectra were determined as previously reported [22]. The NMR spectra were

acquired at 300.13 ($^1$H) and 75.48 ($^{13}$C) MHz, in CDCl$_3$. The chemical shifts are informed in the $\delta$ scale, in parts per million downfield from TMS, employed as the internal standard. The high-resolution mass spectra were acquired on a Bruker MicroTOF-Q II spectrometer (UMYMFOR, Buenos Aires, Argentina). The ions were detected by electrospray ionization, under the positive ion mode.

## 2.3. General procedure for the syntheses of the methoximes

The starting aldehyde or ketone (0.30 mmol) was transferred to the test tube without special protection against air (O$_2$) and dissolved with stirring in absolute EtOH (2.5 ml). Anhydrous NaOAc (37 mg, 0.45 mmol, 1.5 equiv.) and MeONH$_2$·HCl (37.5 mg, 0.45 mmol, 1.5 equiv.) were added and the system was treated with MnCl$_2$·4H$_2$O (3 mg, 5 mol%).

The mixture was warmed to 50°C and TLC was employed to keep track of the evolution of the process. When the reaction reached completion, it was diluted with brine (10 ml) and the products were extracted with EtOAc (3 × 10 ml). The combined organic layers were dried (MgSO$_4$) and the solvent was removed under reduced pressure. The residue was purified by column chromatography under gradient of solvent polarity conditions.

**(E)-2-Methoxybenzaldehyde O-methyloxime (1) [28]**

Colourless oil; yield: 81%. IR (film, $\hat{\upsilon}$): 3075, 3003, 2961, 2938, 2899, 2837, 2816, 1607, 1601, 1487, 1464, 1456, 1439, 1341, 1302, 1285, 1252, 1161, 1111, 1055, 1028, 920, 853 and 754 cm$^{-1}$. $^1$H NMR: $\delta$ = 8.45 (1H, s, H-1), 7.78 (1H, dd, $J$ = 1.7 and 7.8, H-6′), 7.33 (1H, dt, $J$ = 1.7 and 8.0, H-4′), 6.94 (1H, t, $J$ = 8.0, H-5′) and 6.88 (1H, d, $J$ = 8.0, H-3′), 3.96 (3H, s, N-OMe), 3.83 (s, 3H, C2′-OMe). $^{13}$C NMR: $\delta$ = 157.5 (C-2′), 144.8 (C-1), 131.1 (C-4′), 126.4 (C-6′), 120.8 (C-1′ and C-5′), 111.0 (C-3′), 61.9 (N-OMe) and 55.6 (OMe).

**(E)-2-Bromobenzaldehyde O-methyl oxime (2) [28,29]**

Yellow oil; yield: 98%. $^1$H NMR: $\delta$ = 8.45 (1H, s, H-1), 7.87 (1H, dd, $J$ = 1.8 and 7.7. H-6′), 7.55 (1H, dd, $J$ = 1.4 and 7.7, H-3′), 7.30 (1H, dt, $J$ = 1.4 and 7.7, H-5), 7.21 (1H, dt, $J$ = 1.8 and 7.7, H-4′), 4.00 (3H, s, N-OMe). $^{13}$C NMR: $\delta$ = 147.9 (C-1), 133.1 (C-3′), 131.5 (C-1′), 131.0 (C-4′), 127.5 (C-6′), 127.5 (C-5′), 123.8 (C-2′), 62.3 (N-OMe).

**(E)-2-Chlorobenzaldehyde O-methyloxime (3) [28]**

Yellow oil; yield: 97%. IR (film, $\hat{\upsilon}$): 2936, 1601, 1472, 1433, 1342, 1209, 1063, 1045, 924, 851, 754, 704, 619 and 602 cm$^{-1}$. $^1$H NMR: $\delta$ = 8.48 (1H, s, H-1), 7.88 (1H, dd, $J$ = 2.2 and 7.3, H-6′), 7.38–7.21 (3H, m, H-3′, H-4′ and H-5′) and 3.99 (3H, s, N-OMe). $^{13}$C NMR: $\delta$ = 145.6 (C-1), 135.1 (C-2′), 133.8 (C-5′), 130.7 (C-1′), 129.8 (C-4′), 127.1 (C-6′), 126.9 (C-3′) and 62.2 (N-OMe).

**(E)-2-(Trifluoromethyl)benzaldehyde O-methyloxime (4) [22]**

Colourless oil; yield: 68%. IR (film, $\hat{\upsilon}$): 2968, 2941, 2903, 2822, 1487, 1454, 1360, 1315, 1283, 1171, 1125, 1067, 1049, 1034, 962, 932, 854, 768, 750, 662 and 629 cm$^{-1}$. $^1$H NMR: $\delta$ = 8.43 (1H, q, $J$ = 2.1, H-1), 8.06 (1H, d, $J$ = 7.8, H-6′), 7.67 (1H, d, $J$ = 7.8, H-3′), 7.54 (1H, t, $J$ = 7.5, H-5′*), 7.46 (1H, t, $J$ = 7.5, H-4′*) and 4.02 (3H, s, N-OMe). $^{13}$C NMR: $\delta$ = 145.2 (C-1), 131.9 (C-5′), 130.4 (C-1′), 129.4 (C-4′), 128.4 (C-2′), 128.2 (q, $J$ = 30.8, CF$_3$), 127.2 (C-6′), 125.8 (q, $J$ = 5.5, C-3′) and 62.3 (N-OMe). EI-MS (m/z, %): 203 (M$^+$, 89), 152 (77), 145 (100), 125 (26) and 75 (51). HRMS (ESI-TOF): m/z [M + H]$^+$ calcd. for C$_9$H$_9$F$_3$NO: 204.0611; found: 204.0608.

**(E)-3,4-Dihydroxybenzaldehyde O-methyloxime (5) [30]**

Yellow oil; yield: 94%. $^1$H NMR: $\delta$ = 7.95 (1H, s, H-1), 7.19 (1H, d, $J$ = 2.0, H-2′), 6.96 (1H, dd, $J$ = 2.0 and 8.2, H-6′), 6.86 (1H, d, $J$ = 8.2, H-5′), 5.60 (br s, 2H, OH-3′ and OH-4′) and 3.94 (3H, s, N-OMe). $^{13}$C NMR: $\delta$ = 148.5 (C-1), 145.7 (C-4′*), 143.6 (C-3′*), 125.2 (C-1′), 121.7 (H-6′), 115.4 (H-5′), 112.9 (C-2′) and 61.8 (N-OMe).

**(E)-3,4-Dimethoxybenzaldehyde O-methyloxime (6) [31]**

White solid; yield: 94%; mp 56–57°C. $^1$H NMR: $\delta$ = 8.00 (1H, s, H-1), 7.25 (1H, d, $J$ = 2.0, H-2′), 7.01 (1H, dd, $J$ = 2.0 and 8.2, H-6′), 6.84 (1H, d, $J$ = 8.2, H-5′), 3.96 (3H, s, N-OMe), 3.93 (3H, s, 4′-OMe*) and 3.90 (3H, s, 3′-OMe*). $^{13}$C NMR: $\delta$ = 150.7 (C-3′*), 149.3 (C-4′*), 148.4 (C-1), 125.1 (C-1′), 121.6 (C-6′), 110.7 (C-5′), 108.0 (C-2′), 61.9 (N-OMe) and 55.9 (3′-OMe and 4′-OMe).

**(E)-2,4,6-Trimethoxybenzaldehyde O-methyloxime (7)**

White solid; yield: 97%; mp 82–84°C. IR (KBr, $\hat{\upsilon}$): 3566, 3545, 3481, 3420, 3416, 2999, 2941, 2901, 2843, 2820, 2365, 2344, 1614, 1599, 1570, 1474, 1458, 1339, 1227, 1202, 1155, 1126, 1067, 1053, 1030 and 810 cm$^{-1}$. $^1$H NMR: $\delta$ = 8.41 (1H, s, H-1), 6.13 (2H, s, H-3′ and H-5′), 3.97 (3H, s, N-OMe), 3.84 (6H, s, 2′-OMe and 6′OMe) and 3.83 (3H, s, 4′-OMe). $^{13}$C NMR: $\delta$ = 162.4 (C-2′ and C-6′), 160.1 (C-4′*), 144.4 (C-1), 102.4 (C-1), 90.7 (C-3′ and C-5′), 61.5 (N-OMe), 56.0 (2′-OMe and 6′-OMe) and 55.3 (4′-OMe′). HRMS (ESI-TOF): m/z [M + H]$^+$ calcd. for C$_{11}$H$_{16}$NO$_4$: 226.1074; found: 226.1075.

### (E)-2-Bromo-5-methoxybenzaldehyde O-methyloxime (8)

Yellow oil; yield: 87%. IR (film, $\hat{\upsilon}$): 3003, 2963, 2936, 2899, 2833, 2818, 1591, 1564, 1464, 1418, 1404, 1348, 1292, 1273, 1233, 1171, 1061, 1047, 1016 and 910 cm$^{-1}$. $^1$H NMR: $\delta$ = 8.41 (1H, s, H-1), 7.42 (1H, d, $J$ = 8.9, H-6'), 7.39 (1H, d, $J$ = 3.1, H-3'), 6.81 (1H, dd, $J$ = 3.1 and 8.8, H-4'), 4.00 (3H, s, N-OMe) and 3.83 (3H, s, OMe). $^{13}$C NMR: $\delta$ = 158.9 (C-5'), 148.0 (C-1), 133.7 (C-3'), 132.1 (C-1'), 118.5 (C-4'), 114.6 (C-2'), 111.2 (C-6'), 62.3 (N-OMe) and 55.6 (OMe). HRMS (ESI-TOF): $m/z$ [M + H]$^+$ calcd. for C$_9$H$_{11}$BrNO$_2$: 243.9968; found: 243.9970.

### (E)-2,5-Dimethoxybenzaldehyde O-methyloxime (9) [32]

Yellow oil; yield: 83%. IR (film, $\hat{\upsilon}$): 3001, 2940, 2901, 2835, 1506, 1495, 1456, 1339, 1221, 1053 and 908 cm$^{-1}$. $^1$H NMR: $\delta$ = 8.42 (1H, s, C-1), 7.33 (1H, d, $J$ = 3.1, H-6'), 6.90 (1H, dd, $J$ = 3.1 and 9.0, H-4'), 6.82 (1H, d, $J$ = 9.0, H-3'), 3.97 (3H, s, N-OMe), 3.80 (3H, s, OMe*) and 3.79 (3H, s, OMe*). $^{13}$C NMR: $\delta$ = 153.7 (C-2'*), 152.1 (C-5'*), 144.6 (C-1), 121.3 (C-1'), 117.6 (C-4'), 112.8 (H-3'), 110.1 (C-6'), 61.9 (N-OMe), 56.3 (2-OMe*) and 55.8 (5-OMe*).

### (E)-1-(4-Methoxyphenyl)ethan-1-one O-methyloxime (10) [33]

White solid; yield: 84%; mp 54–55°C. IR (KBr, $\hat{\upsilon}$): 3414, 1616, 1522, 1258, 1049, 891 and 827 cm$^{-1}$. $^1$H NMR: $\delta$ = 7.60 (2H, dd, $J$ = 2.2 and 6.8, H-2' and H-6'), 6.89 (2H, dd, $J$ = 2.2 and 6.8, H-3' and H-5'), 3.98 (3H, s, N-OMe), 3.82 (3H, s, OMe) and 2.20 (3H, s, H-2). $^{13}$C NMR: $\delta$ = 160.4 (C-4), 154.2 (C-1), 129.2 (C-1'), 127.4 (C-2'and C-6'), 113.8 (C-3'and C-5'), 61.8 (N-OMe), 55.3 (OMe) and 12.5 (C-2).

### (E)-1-(p-Tolyl)ethan-1-one O-methyloxime (11) [34]

Colourless oil; yield: 77%. $^1$H NMR: $\delta$ = 7.54 (2H, d, $J$ = 8.1, H-2' and H-6'), 7.17 (2H, d, $J$ = 7.9, H-3' and H-5'), 3.98 (3H, s, N-OMe), 2.36 (3H, s, 4'-Me) and 2.21 (3H, s, H-2). $^{13}$C NMR: $\delta$ = 154.6 (C-1), 139.0 (C-4'), 133.8 (C-1'), 129.1 (C-3' and C-5'), 125.9 (C-2' and C-6'), 61.8 (N-OMe), 21.2 (4'-Me) and 12.6 (C-2').

### (E)-1-(4-Bromophenyl)ethan-1-one O-methyloxime (12) [22,33]

Colourless oil (isomeric mixture $E/Z$ = 87:13); yield: 93%. IR (film, $\hat{\upsilon}$): 2936, 1609, 1587, 1485, 1395, 1317, 1084, 1049, 1009, 895 and 824 cm$^{-1}$. $^1$H NMR: $\delta$ = 7.55–7.46 (4H, m, H2', H-3', H-5' and H-6'), 3.99 (3H, s, N-OMe) and 2.19 (3H, s, H-2). $^{13}$C NMR: $\delta$ = 153.5 (C-1), 135.5 (C-1'), 131.5 (C-3' and C-5'), 127.6 (C-2' and C-6'), 123.3 (C-4'), 62.0 (N-OMe) and 12.4 (C-2).

### (E)-1-(4-Fluorophenyl)ethan-1-one O-methyloxime (13) [33,34]

Yellowish oil (mixture $E/Z$ = 94:6); yield: 61%. IR (film, $\hat{\upsilon}$): 2959, 2926, 2853, 2818, 1730, 1603, 1512, 1369, 1315, 1233, 1159, 1084, 1051, 1015, 895, 835, 575, 444, 436, 430, 411 and 403 cm$^{-1}$. $^1$H NMR: $\delta$ = 7.67–7.59 (2H, m, H-2' and H-6'), 7.09–7.00 (2H, m, H-3' and H-5'), 3.99 (3H, s, N-OMe) and 2.21 (3H, s, H-2). $^{13}$C NMR: $\delta$ = 163.3 ($J$ = 248.6, C-4'), 153.6 (C-1), 132.8 (C-1'), 127.9 (C-2'*), 127.8 (C-6'*), 115.5 (C-3'*), 115.2 (C-4'*), 61.9 (N-OMe) and 12.6 (C-2).

### (E)- and (Z)-1-(3,4-Dichlorophenyl)ethan-1-one O-methyloxime (14) [35]

White solid (mixture $E/Z$ = 85:15); yield: 86%; mp 39–41°C. IR (KBr, $\hat{\upsilon}$): 3005, 2938, 2899, 2820, 2374, 2345, 1751, 1734, 1653, 1541, 1472, 1375, 1315, 1140, 1051 and 1028 cm$^{-1}$.

*E*-isomer: $^1$H NMR: $\delta$ = 7.75 (1H, d, $J$ = 1.9, H-2'), 7.49 (1H, dd, $J$ = 1.9 and 8.4, H-6'), 7.42 (1H, d, $J$ = 8.4, H-5'), 4.00 (3H, s, N-OMe) and 2.18 (3H, s, H-2). $^{13}$C NMR: $\delta$ = 152.3 (C-1), 136.5 (C-1'), 133.0 (C-3'*), 132.7 (C-4'*), 130.3 (C-5'), 127.8 (C-2'), 125.1 (C-6'), 62.2 (N-OMe) and 12.2 (C-2).

*Z*-isomer: $^1$H NMR: $\delta$ = 7.61 (1H, d, $J$ = 1.6, H-2'), 7.43 (1H, d, $J$ = 8.4 Hz, H-5'), 7.33 (1H, dd, $J$ = 1.7 and 8.4, H-6'), 3.86 (3H, s, N-OMe) and 2.17 (3H, s, H-2). $^{13}$C NMR: $\delta$ = 151.1 (C-1), 136.5 (C-1'), 133.9 (C-3'*), 132.4 (C-4'*), 130.1 (C-2'*), 130.1 (C-5'*), 127.4 (C-6'), 61.9 (N-OMe) and 21.3 (C-2).

### (E)-1-(3,4-Dimethoxyphenyl)ethan-1-one O-methyloxime (15) [36]

White solid; yield: 96%; mp 58–59°C. IR (KBr, $\hat{\upsilon}$): 2961, 1576, 1516, 1464, 1416, 1339, 1277, 1252, 1231, 1175, 1150, 1045, 1020, 918, 866 and 812 cm$^{-1}$. $^1$H NMR: $\delta$ = 7.30 (1H, d, $J$ = 2.0, H-2'), 7.14 (1H, dd, $J$ = 2.1 and 8.4, H-6'), 6.85 (1H, d, $J$ = 8.5, H-5'), 3.99 (3H, s, N-OMe), 3.93 (3H, s, 4'-OMe*), 3.90 (3H, s, 3'-OMe*) and 2.21 (3H, s, H-2). $^{13}$C NMR: $\delta$ = 154.2 (N-OMe), 150.0 (C-4'), 148.9 (C-3'), 129.4 (C-1'), 119.1 (C-6'), 110.6 (C-5'), 108.6 (C-2'), 61.8 (N-OMe), 55.9 (3'-OMe and 4'-OMe) and 12.5 (C-2).

### (E)-1-(2-Nitrophenyl)ethan-1-one O-methyloxime (16) [37]

Yellow oil (isomeric mixture $E/Z$ = 77:23); yield: 89%. IR (film, $\hat{\upsilon}$): 2938, 2820, 1611, 1537, 1346, 1047, 893 and 787 cm$^{-1}$. $^1$H NMR: $\delta$ = 8.00 (1H, dd, $J$ = 1.1 and 8.0, H-3'), 7.64 (1H, dt, $J$ = 1.4 and 7.5, H-5'), 7.53 (1H, dt, $J$ = 1.5 and 7.8, H-4'), 7.46 (1H, dd, $J$ = 1.4 and 7.6, H-6'), 3.96 (3H, s) and 2.17 (3H, s). $^{13}$C NMR: $\delta$ = 154.2 (C-1), 148.0 (C-2'), 133.6 (C-1'), 133.3 (C-6'), 130.6 (C-4'), 129.5 (C-5'), 124.6 (C-3'), 62.1 (N-OMe) and 16.0 (C-2).

### (E)-1-(2-Hydroxy-4-methoxyphenyl)ethan-1-one O-methyloxime (17) [38]

White solid; yield: 97%; mp 52–54°C. $^1$H NMR: $\delta$ = 11.49 (1H, s, OH), 7.29 (1H, d, $J$ = 8.8, H-6'), 6.49 (1H, d, $J$ = 2.7, H-3'), 6.45 (1H, dd, $J$ = 2.7 and 8.8, H-5'), 3.97 (3H, s, N-OMe), 3.80 (3H, s, OMe) and 2.26

(3H, s, H-2). $^{13}$C NMR: $\delta$ = 161.7 (C-4′*), 159.6 (C-2′*), 158.1 (C-1), 128.6 (C-6′), 111.7 (C-1′), 106.2 (C-5′), 101.6 (C-3′), 62.2 (N-OMe), 55.3 (OMe), 11.4 (C-2).

### (E)-1-(2-Hydroxyphenyl)ethan-1-one O-methyloxime (18) [32]

Yellow oil; yield: 92%. IR (film, ṽ): 3003, 2965, 2938, 2899, 2820, 2376, 2349, 2318, 2311, 1620, 1607, 1506, 1456, 1300, 1250, 1053 and 920 cm$^{-1}$. $^{1}$H NMR: $\delta$ = 11.27 (1H, s, OH), 7.41 (1H, dd, $J$ = 1.5 and 8.0, H-6′), 7.25 (1H, td, $J$ = 4.2 and 8.5, H-4′), 6.97 (1H, dd, $J$ = 1.0 and 8.2, H-3′), 6.89 (1H, td, $J$ = 4.1 and 8.2, H-5′), 4.00 (3H, s, N-OMe), 2.31 (3H, s, H-2). $^{13}$C NMR: $\delta$ = 158.3 (C-2′), 157.9 (C-1), 130.7 (C-4′), 127.5 (C-6′), 119.0 (C-5′), 118.3 (C-1′), 117.3 (C-3′), 62.4 (N-OMe), 11.4 (C-2).

### 4-[(1E)-N-Methoxypropanimidoyl]benzene-1,3-diol (19) [39]

Yellowish oil; yield: 86%. IR (film, ṽ): 3379, 2978, 2938, 1703, 1634, 1614, 1520, 1454, 1250, 1047, 970, 891, 851 and 743 cm$^{-1}$. $^{1}$H NMR: $\delta$ = 11.73 (1H, s, OH), 7.27 (1H, d, $J$ = 8.5, H-5′), 6.46 (1H, d, $J$ = 2.5, H-2′), 6.41 (1H, dd, $J$ = 2.6 and 8.6, H-4′), 5.67 (br s, 1H, OH), 3.96 (3H, s, N-OMe), 2.77 (2H, q, $J$ = 7.6, H-2) and 1.17 (3H, t, $J$ = 7.6, H-3). $^{13}$C NMR: $\delta$ = 163.2 (C-1), 160.0 (C-1′), 157.9 (C-3′), 128.8 (C-5′), 110.7 (C-6′), 107.1 (C-4′), 103.9 (C-2′), 62.3 (N-OMe), 18.8 (Me) and 11.4 (C-3). EI-MS ($m/z$, %): 195 (M$^{+}$, 83), 180 [(M-15)$^{+}$, 1], 164 (33), 135 (100) and 108 (57).

### (E)-1-(4-Methoxyphenyl) propan-1-one O-methyloxime (20) [32]

Colourless oil; yield: 88%. $^{1}$H NMR: $\delta$ = 7.58 (2H, dd, $J$ = 2.2 and 6.9, H-2′ and H-6′), 6.89 (2H, dd, $J$ = 2.2, 6.9, H-3′ and H-5′), 3.96 (3H, s, N-OMe), 3.82 (3H, s, OMe), 2.72 (2H, q, $J$ = 7.6, H-2) and 1.12 (3H, t, $J$ = 7.6, H-3). $^{13}$C NMR: $\delta$ = 160.3 (C-1), 159.4 (C-4′), 128.1 (C-1′), 127.6 (C-2′ and C-6′), 113.8 (C-3′ and C-5′), 61.7 (N-OMe), 55.3 (OMe), 20.0 (C-2) and 11.2 (C-3).

### (S,E)-4-(prop-1-en-2-yl)cyclohex-1-ene-1-carbaldehyde O-methyl oxime (21) [22]

Colourless oil; yield: 80% (greater than 98% $E$). $[\alpha]_D^{16}$ = –134.4 ($c$, 0.84, CHCl$_3$). IR (film, ṽ): 2936, 2899, 1643, 1454, 1435, 1373, 1179, 1059, 1045, 947, 897 and 667 cm$^{-1}$. $^{1}$H NMR: $\delta$ = 7.65 (1H, s, H-1), 5.98 (1H, bdd, $J$ = 2.4, 5.0, H-2′), 4.75 (1H, bs, H-1″b), 4.72 (1H, bs, H-1″a), 3.86 (3H, s, N-OMe), 2.47 (1H, ddd, $J$ = 2.8, 2.8 and 17.4, H-3′$_{eq}$), 2.37–2.15 (3H, m, H-3′$_{ax}$ and H-6′), 2.15–2.01 (1H, m, H-4′), 1.93–1.83 (1H, m, H-5′$_{eq}$), 1.75 (3H, s, H-3″) and 1.48 (dddd, 1H, $J$ = 5.4, 11.2, 12.4 and 12.8, H-5′$_{ax}$). $^{13}$C NMR: $\delta$ = 151.6 (C-1), 149.2 (C-2″), 134.9 (C-2′), 132.6 (C-1′), 109.0 (C-1″), 61.6 (N-OMe), 40.9 C-4′, 31.3 (C-3′), 26.8 C-6′, 23.9 (C-5′) and 20.7 (C-3″). EI-MS ($m/z$, %): 179 (M$^{+}$, 2), 164 [(M-15)$^{+}$, 2], 138 (4), 110 (50), 80 (100) and 68 (44).

### (2R,5R,E)-2-Methyl-5-(prop-1-en-2-yl)cyclohexan-1-one O-methyl oxime (22) [22]

Colourless oil; yield: 81% (greater than 95% $E$). $[\alpha]_D^{16}$ = – 86.0 ($c$, 0.74, CHCl$_3$). IR (film, ṽ): 2963, 2930, 2855, 1643, 1447, 1378, 1261, 1051, 905, 880 and 849 cm$^{-1}$. $^{1}$H NMR: $\delta$ = 4.73 (2H, bs, $w_{1/2}$ = 3.0, H-1′), 3.82 (3H, s, N-OMe), 3.32 (1H, ddd, $J$ = 2.0, 3.8 and 13.5, H-6), 2.24–2.11 (1H, m, H-2), 2.05 (1H, tdd, $J$ = 3.6, 11.8 and 11.8), 1.99–1.90 (1H, m, H-4$_{eq}$), 1.90–1.79 (1H, m, H-3$_{eq}$), 1.73 (3H, s, H-3′), 1.58 (1H, t, $J$ = 13.0, H-6$_{eq}$), 1.41 (1H, ddd, $J$ = 3.3, 12.7 and 24.5, H-3$_{ax}$), 1.24 (1H, ddd, $J$ = 3.3, 12.7 and 24.5, H-4$_{ax}$), 1.10 (3H, d, $J$ = 6.4, Me-2). $^{13}$C NMR: $\delta$ = 161.7 (C-1), 148.7 (C-2′), 109.2 (C-1′), 61.1 (N-OMe), 44.9 (C-5), 37.2 (C-2), 35.4 (C-4), 30.9 (C-3), 29.9 (C-6), 20.8 (C-3′) and 16.4 (Me-2). EI-MS ($m/z$, %): 181 (M$^{+}$, 2), 166 [(M-15)$^{+}$, 2], 149 (2), 125 (8), 109 (13), 97 (50) and 71 (100).

## 3. Results and discussion

At the beginning of the research, we defined initial conditions for the transformation on the basis of past experience. NaOAc was employed to remove HCl and set free the methoxylamine. In our previous investigation on CeCl$_3$.7H$_2$O as a catalyst, we detected that small amounts of water appeared to slightly reduce the reaction rate, not being an obstacle for the transformation [22]. After confirming this observation, the anhydrous form of the salt was used for the experiments. NaOAc is a GRAS (Generally Regarded as Safe) substance, which helps to fulfil our aim of process eco-friendliness [40]. The suitability of different manganese salts, as well as the proper reaction solvent and temperature, were then confirmed, monitoring by TLC (run every 10 min) the time required for the exhaustion of the starting carbonyl compound.

As shown in table 1, when the reactions were performed in absolute EtOH, it was observed that in the absence of manganous salts, the starting ketone was not fully consumed even after 120 min (Entry 1). Considering that the catalytic ability of simple manganous salts also depends on the counterion [41,42], the effect of 5 mol% of different manganous salts such as the sulfate (Entry 2), acetate (Entry 3), perchlorate (Entry 4) and MnCl$_2$.4H$_2$O (Entry 5), was examined.

Although it may be argued that the addition of NaOAc to the reaction may afford Mn(OAc)$_2$, it was found that the counterion seems to be relevant, since MnCl$_2$.4H$_2$O turned the transformation most

**Table 1.** Optimization of the conditions for the methoximation.

| entry no. | solvent | source of Mn(II)[a] | temperature (°C) | time (min)[b] |
|---|---|---|---|---|
| 1 | EtOH | — | 50 | >120 |
| 2 | EtOH | $MnSO_4.H_2O$ | 50 | 45 |
| 3 | EtOH | $Mn(OAc)_2.4H_2O$ | 50 | 60 |
| 4 | EtOH | $Mn(ClO_4)_2.6H_2O$ | 50 | 60 |
| 5 | EtOH | $MnCl_2.4H_2O$ | 50 | 30 |
| 6 | MeOH | $MnCl_2.4H_2O$ | 50 | >75 |
| 7 | 2-PrOH | $MnCl_2.4H_2O$ | 50 | >75 |
| 8 | EtOH | $MnCl_2.4H_2O$ | 27 | >60 |
| 9 | EtOH | $MnCl_2.4H_2O$ | 40 | 50 |

[a]A level of 5 mol% of the catalyst was used.
[b]Time to achieve complete consumption of the starting material, according to the TLC.

efficiently, resulting in full depletion of the starting ketone in only 30 min (Entry 5 versus Entry 3). Therefore, the latter was employed for further studies.

$MnCl_2.4H_2O$ is also a GRAS substance [43], suitable for use in protic solvents. This mild and moisture compatible Lewis acid has been found to promote a handful of relevant transformations, such as mesylate to chloride conversion [44], esterification and transesterification reactions [45,46], Michael addition [47], glucose to fructose isomerization [48], condensation of carbonyls with amine derivatives [49], carbonyl reduction [50] and the reductive amination of ketones [51]. The salt has also been employed as a catalyst for cross couplings leading to C–X (X=N, S) [52,53] or C–C bond formation [54,55], and in C–H activation reactions resulting in further C–C bond formation [56–59]. $MnCl_2.4H_2O$ was also supported on montmorillonite K-10, and used in the catalytic synthesis of benzaldehyde derivatives, by the $H_2O_2$-mediated oxidation of the corresponding alcohols [60].

Once the base, as well as the promoter and its level were defined, the performances of MeOH (Entry 6), and 2-PrOH (Entry 7) were compared against the use of EtOH (Entry 5) as reaction medium. It was verified that in the latter case the transformation concluded in just 30 min, whereas the transformation was slower in the other media, not reaching completeness even after 75 min. This observation enabled us to conclude that EtOH should be the solvent of choice. Fortunately, this is the less expensive and less toxic one among the tested solvents.

Finally, comparing the methoximations run in EtOH under $MnCl_2.4H_2O$ catalysis at 27°C (Entry 8), 40°C (Entry 9) and 50°C (Entry 5) revealed that the rate of the transformation was markedly slowed as the system approached room temperature, and confirmed that 50°C was a suitable reaction condition, still being mild when compared with methods requiring solvent reflux (see electronic supplementary material, table S1).

Once the reaction conditions for optimum efficiency were established, and it was demonstrated that the transformation is efficient under low catalyst load and without the need for addition of external ligands, we began the exploration of the scope of the method, by submitting to methoxime formation an array of aldehydes and ketones bearing different steric and electronic characteristics (table 2). In order to better highlight the advantages of the proposed catalyst, the preparation of new as well as known compounds was performed.

In general, it was observed that the transformation was efficient, proceeding in over 75% yield, except for the fluorinated compounds (Entries 4 and 13). The aldehydes (Entries 1–9) were more reactive than the ketones, and the reactions were generally completed in 5 min. Furthermore, the process seemed to be insensitive to steric effects, since the yields of the methoximes of *ortho*-substituted aldehydes (Entries 1–4) or those carrying additional functionalities at other positions (Entries 7–9) were reasonably similar to those arising from aldehydes lacking *ortho*-substituents (Entries 5 and 6).

**Table 2.** Scope of the $MnCl_2.4H_2O$-catalysed methoximation of aromatic aldehydes and ketones.

Reaction conditions: $MeONH_2.HCl$, NaOAc, $MnCl_2.4H_2O$, EtOH, 50°C

| entry/ prod. no. | methoxime | time (min) | yield (%) | entry/ prod. no. | methoxime | time (min) | yield (%) | entry/ prod. no. | methoxime | time (min) | yield (%) |
|---|---|---|---|---|---|---|---|---|---|---|---|
| 1 | | 5 | 81 | 9 | | 15 | 83 | 17 | | 480 | 97 |
| 2 | | 5 | 98 | 10 | | 30 | 84 | 18 | | 75 | 92 |
| 3 | | 5 | 97 | 11 | | 15 | 77 | 19 | | 2100 | 86 |
| 4 | | 5 | 68 | 12 | | 20 | 93 | 20 | | 75 | 88 |
| 5 | | <1 | 94 | 13 | | 15 | 61 | 21 | | 15 | 80 |
| 6 | | 5 | 94 | 14 | | 18 | 86 | 22 | | 35 | 81 |
| 7 | | 5 | 97 | 15 | | 18 | 96 | | | | |
| 8 | | 5 | 87 | 16 | | 360 | 89 | | | | |

**Scheme 2.** Proposed reaction mechanism for the MnCl$_2$.4H$_2$O-catalysed methoximation.

On the other hand, the results evidenced that the ketones lacking an *ortho*-substituent (Entries 10–15) afforded the corresponding products in 15–30 min, while some of their congeners with an *ortho*-functionalization (Entries 16–19) reacted at a pace which depended on the nature of the substituent.

An *ortho*-nitro group (Entry 16) caused a reduced reaction rate, and the same outcome was observed with the presence of an *ortho*-OH substituent (Entries 17–19). This effect may be a result of the known hydrogen bonding capability of the OH group with the nearby carbonyl [61,62]. Interestingly, among these phenolic derivatives, compounds displaying an additional oxygen functionality (OMe, OH, Entries 17 and 19) exhibited a markedly slower reaction rate.

Finally, the effect of steric hindrance on the side chain was examined (Entries 19 and 20). It was detected that the reaction afforded highly satisfactory yields of the expected products (86–88%); however, it took longer (Entry 20 versus Entry 10), especially in the case of Entry 19, where the ketone displays an *ortho*-OH group.

Interestingly, in most of the cases, signs of E/Z (*anti/syn*) mixtures of methoximes (with one of them clearly prevailing) were easily visible in the NMR spectra of the compounds. Unfortunately, however, these mixtures were not chromatographically separable on a silica gel column.

In addition, we confirmed that the reaction has a wide scope, observing that the conditions are also useful for efficiently accessing methoximes derived from non-aromatic substrates, as exemplified with the derivatives of perillaldehyde (Entry 21) and dihydrocarvone (Entry 22). Further, thanks to the very mild reaction conditions, no signs of the potentially competing Michael addition were observed. All these reaction features suggest that, contrary to previous beliefs [19], the synthetic potential of this simple and straightforward metal-mediated approach may be high compared to the state of the art (see electronic supplementary material, table S1).

With all the data in our hands, a comparison between our previous results with CeCl$_3$.7H$_2$O as catalyst and the proposed methodology employing MnCl$_2$.4H$_2$O as promoter (see electronic supplementary material, table S2) was carried out on a set of eight compounds, including aromatic aldehydes and ketones, as well as a couple of alicyclic derivatives. The analysis revealed that MnCl$_2$.4H$_2$O displays a similar and often even better performance than the cerium(III) salt, and that it can conveniently act as its alternative or substitute under analogous reaction conditions. Furthermore,

advantageously, the manganese(II) salt is a GRAS compound, being easily available and less costly than $CeCl_3.7H_2O$.

At the present stage, the intimate details of the reaction mechanism are not clear. However, scheme 2 illustrates a stepwise inner-works speculation, based on the classical mechanism and literature precedents [63,64]. It is assumed that the coordination of $MnCl_2$ to the oxygen of the carbonyl motif of model compound **A** triggers the transformation, yielding the activated intermediate (*i*) and releasing chloride anion.

Plausibly, the extent of the coordination depends on the nature of the counteranion in the Mn(II) Lewis acid catalyst, which would explain the differences in reactivity between $MnSO_4$ and $MnCl_2$.

In turn, this intermediate (*i*) can suffer a nucleophilic attack by the nitrogen atom of methoxylamine, which would result in the tetrahedral intermediate (*ii*), *O*- and *N*-coordinated to the manganese centre, able to undergo an intramolecular proton transfer to afford the protonated hemiaminal-type intermediate (*iii*), analogous to that reported during the titanium(IV)-mediated preparation of oxime ethers [20]. The latter intermediate can next be involved in a chloride-assisted deprotonation, which should result in dehydration and regeneration of the catalyst, with concomitant release of the methoxime product (**B**).

It is very likely that all the steps from (**A**) to the last intermediate (*iii*) are reversible, unlike the step from intermediate (*iii*) toward (**B**), which under similar reaction conditions is also the rate-limiting step of the sequence [65–67]. However, once dehydration took place and compound (**B**) was formed, it is less likely that the latter could be hydrolyzed to the starting material (**A**). The validity of this conjecture stems from the fact that oxime ethers are relatively stable to hydrolysis [68,69] and that the salt is unable to catalyse deoximation [70] under the reaction conditions.

# 4. Conclusion

In order to fulfil the objective need of mild and environmentally conscious catalytic methoximation alternatives, a manganese(II)-catalysed reaction for aromatic aldehydes and ketones based on the easily available $MnCl_2.4H_2O$ reagent was developed as a novel and original application of this salt. This metal-catalysed transformation proved to have considerable substrate scope; further, it does not require the addition of an external ligand and features a low catalyst loading.

The reaction evidences a unique and still unreported catalytic application of this scarcely employed manganese salt. This derivatization of carbonyl compounds can be efficiently performed with a minimalistic setup, under mild conditions and in short time, being a suitable alternative to methoximation processes which so far have been carried out in generally harsher reaction scenarios and/or longer times, usually employing less convenient reagents. These features advance the state of the art of the methoximation reaction; additionally, it should be considered that due to the marked abundance of manganese on Earth and the eco-friendliness of $MnCl_2.4H_2O$, the catalyst is also a competent, highly convenient and more sustainable substitute of the previously reported $CeCl_3.7H_2O$. These characteristics suggest that the proposed approach is a promising protocol with potentially high synthetic value, that avoids the use of high temperature, costly or undesirable solvents, excess reagents and/or more expensive or even toxic additives. The currently disclosed findings may also be conceived as the starting point for the development of other Mn-based catalysts for this reaction.

Ethics. This research did not require the intervention of the Ethics Committee.

Data accessibility. The figures and tables include all data employed in this research. The authors have uploaded the datasets which support this article as the electronic supplementary material. All data used in this research are included in figures and tables. The datasets supporting this article have been uploaded as part of the electronic supplementary material.

Authors' contributions. A.B.J.B. and T.S.K. designed the study. I.C. and M.C.O. run the experiments and prepared samples for spectroscopic analysis and performed spectroscopic analyses. T.S.K., A.B.J.B. and I.C. accumulated and compiled the data and proceeded to their analyses. All the authors participated in the interpretation and discussion of the results; they also took part in elaborating the drafts, performing the revision and approving the manuscript in its final form.

Competing interests. We declare we have no competing interests.

Funding. Financial support came from the Secretaría de Ciencia y Tecnología de la UNR (Project BIO457), Consejo Nacional de Investigaciones Científicas y Técnicas (CONICET, PUE IQUIR 2016) and Agencia Nacional de Promoción de la Investigación, el Desarrollo Tecnológico y la Innovación (PICT 2017-0149).

Acknowledgements. M.C.O. is thankful to SECyT-UNR for her internship at the Institute of Chemistry of Rosario (IQUIR, CONICET-UNR) and I.C. acknowledges CONICET for his Fellowship (Doctoral level). IQUIR, where the research leading to this work was conducted, is an official research institute supported by CONICET.

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
