## [Peer Review File · Royal Society Open Science]

Review History

RSOS-201654.R0 (Original submission)

Review form: Reviewer 1

Is the manuscript scientifically sound in its present form?

Yes

Are the interpretations and conclusions justified by the results?

Yes

Is the language acceptable?

Yes

Do you have any ethical concerns with this paper?

No

Have you any concerns about statistical analyses in this paper?

Yes

Recommendation?

Reject

Comments to the Author(s)

The paper reports an eco-friendly methoximation of aromatic aldehydes and ketones using $\text{MnCl}_2 \cdot 4\text{H}_2\text{O}$ as an easily accessible and efficient catalyst. This work collects standard data on the target subjects, but any exciting aspects are not well included. This is just the addition of new one example without conceptual large innovative steps. Most of the presented data are just standard data on what we usually collect in similar research. In addition, it is not very general and deep comparisons. This work may have values as the addition of new data sets for the specialized research community, but do not have large impacts. With these reasons, I do not recommend the publication of this work.

Review form: Reviewer 2**Is the manuscript scientifically sound in its present form?**

Yes

Are the interpretations and conclusions justified by the results?

Yes

Is the language acceptable?

Yes

Do you have any ethical concerns with this paper?

No

Have you any concerns about statistical analyses in this paper?

No

Recommendation?

Accept with minor revision (please list in comments)

Comments to the Author(s)

In this work, a manganese(II)-catalysed methoximation reaction for aromatic aldehydes and ketones based on the easily available $\text{MnCl}_2 \cdot 4\text{H}_2\text{O}$ reagent was developed. The transformation, which is of considerable substrate scope, does not require the addition of an external ligand and features a low catalyst loading. I think this is a good work, providing interesting results and advance in this catalysis field. Thus, I recommend it to publish after minor revisions.

1. Can authors give a comparison with the benchmark catalysts previously reported in literatures ?

Decision letter (RSOS-201654.R0)

Dear Dr Bracca:

Manuscript ID: RSOS-201654

Title: "Eco-friendly methoximation of aromatic aldehydes and ketones using $\text{MnCl}_2 \cdot 4\text{H}_2\text{O}$ as an easily accessible and efficient catalyst"

Thank you for submitting the above manuscript to Royal Society Open Science. Your paper was sent to reviewers and their comments are included at the bottom of this letter. I apologise that this has taken longer than usual.

In view of the concerns raised by the reviewers, the manuscript has been rejected in its current form. However, a new manuscript may be submitted which takes into consideration these comments.

Please note that resubmitting your manuscript does not guarantee eventual acceptance, and that your resubmission will be subject to peer review before a decision is made.

Your resubmitted manuscript should be submitted by 13-Jul-2021. If you are unable to submit by this date please contact the Editorial Office.

On behalf of the Subject Editor Professor Anthony Stace and the Associate Editor Dr Annette Trunschke

REVIEWER(S) REPORTS:

Associate Editor Comments to Author ():

RSC Associate Editor:

Comments to the Author:

There are two conflicting expert opinions, Reviewer 1 recommends rejection due to lack of novelty, Reviewer 2 recommends acceptance after minor revisions. But Reviewer 2 also asks for comparison of the available work with the state of the art. This reinforces my decision to recommend rejection of the manuscript. The authors may resubmit. However, they should then provide a powerful justification of what is new about the paper and what new insights it will provide for catalysis research in general.

RSC Subject Editor:

Comments to the Author:

(There are no comments.)

Reviewers' Comments to Author:

Reviewer: 1

Comments to the Author(s)

The paper reports an eco-friendly methoximation of aromatic aldehydes and ketones using $\text{MnCl}_2 \cdot 4\text{H}_2\text{O}$ as an easily accessible and efficient catalyst. This work collects standard data on the target subjects, but any exciting aspects are not well included. This is just the addition of new one example without conceptual large innovative steps. Most of the presented data are just standard data on what we usually collect in similar research. In addition, it is not very general and deep comparisons. This work may have values as the addition of new data sets for the specialized research community, but do not have large impacts. With these reasons, I do not recommend the publication of this work.

Reviewer: 2

Comments to the Author(s)

In this work, a manganese(II)-catalysed methoximation reaction for aromatic aldehydes and ketones based on the easily available $\text{MnCl}_2 \cdot 4\text{H}_2\text{O}$ reagent was developed. The transformation, which is of considerable substrate scope, does not require the addition of an external ligand and features a low catalyst loading. I think this is a good work, providing interesting results and advance in this catalysis field. Thus, I recommend it to publish after minor revisions.

1. Can authors give a comparison with the benchmark catalysts previously reported in literatures ?

Author's Response to Decision Letter for (RSOS-201654.R0)

See Appendix A.

RSOS-210142.R0

Review form: Reviewer 1

Is the manuscript scientifically sound in its present form?

Yes

Are the interpretations and conclusions justified by the results?

Yes

Is the language acceptable?

Yes

Do you have any ethical concerns with this paper?

No

Have you any concerns about statistical analyses in this paper?

No

Recommendation?

Accept with minor revision (please list in comments)

Comments to the Author(s)

The authors have addressed the novelty of this work compared to previous literature. This work describes the utilization of manganese(II) chloride as a catalyst for methoximation of aromatic aldehydes and ketones. This work can be accepted after the following minor revisions:

1. Why $\text{MnCl}_2 \cdot 4\text{H}_2\text{O}$ provides the best catalytic activity for methoximation of ketones compared to other manganese salts?
2. How about the purity of the manganese chloride salt? Does the purity influence the catalytic activity?
3. Comparison with other catalysts reported in the literature for methoximation of aldehydes and ketones should be provided.

Review form: Reviewer 3

Is the manuscript scientifically sound in its present form?

Yes

Are the interpretations and conclusions justified by the results?

Yes

Is the language acceptable?

Yes

Do you have any ethical concerns with this paper?

Yes

Have you any concerns about statistical analyses in this paper?

Yes

Recommendation?

Accept as is

Comments to the Author(s)

The revised version seems fine.

Decision letter (RSOS-210142.R0)

Dear Dr Bracca:

Title: Eco-friendly methoximation of aromatic aldehydes and ketones using $\text{MnCl}_2 \cdot 4\text{H}_2\text{O}$ as an easily accessible and efficient catalyst

Manuscript ID: RSOS-210142

Thank you for submitting the above manuscript to Royal Society Open Science. On behalf of the Editors and the Royal Society of Chemistry, I am pleased to inform you that your manuscript will be accepted for publication in Royal Society Open Science subject to minor revision in accordance with the referee suggestions. Please find the reviewers' comments at the end of this email. I apologise it has taken longer than usual to be able to send you this decision.

The reviewers and handling editors have recommended publication, but also suggest some minor revisions to your manuscript. Therefore, I invite you to respond to the comments and revise your manuscript.

Because the schedule for publication is very tight, it is a condition of publication that you submit the revised version of your manuscript before 21-May-2021. Please note that the revision deadline will expire at 00.00am on this date. If you do not think you will be able to meet this date please let me know immediately.

1) A text file of the manuscript (tex, txt, rtf, docx or doc), references, tables (including captions) and figure captions. Do not upload a PDF as your "Main Document".

- 2) A separate electronic file of each figure (EPS or print-quality PDF preferred (either format should be produced directly from original creation package), or original software format)
- 3) Included a 100 word media summary of your paper when requested at submission. Please ensure you have entered correct contact details (email, institution and telephone) in your user account
- 4) Included the raw data to support the claims made in your paper. You can either include your data as electronic supplementary material or upload to a repository and include the relevant doi within your manuscript
- 5) All supplementary materials accompanying an accepted article will be treated as in their final form. Note that the Royal Society will neither edit nor typeset supplementary material and it will be hosted as provided. Please ensure that the supplementary material includes the paper details where possible (authors, article title, journal name).

Kind regards,
Dr Laura Smith
Publishing Editor, Journals

On behalf of the Subject Editor Professor Anthony Stace and the Associate Editor Dr Annette Trunschke.

RSC Associate Editor
Comments to the Author:
(There are no comments.)

Reviewer comments to Author:
Reviewer: 1

Comments to the Author(s)

The authors have addressed the novelty of this work compared to previous literature. This work describes the utilization of manganese(II) chloride as a catalyst for methoximation of aromatic aldehydes and ketones. This work can be accepted after the following minor revisions:

1. Why $\text{MnCl}_2 \cdot 4\text{H}_2\text{O}$ provides the best catalytic activity for methoximation of ketones compared to other manganese salts?
2. How about the purity of the manganese chloride salt? Does the purity influence the catalytic activity?
3. Comparison with other catalysts reported in the literature for methoximation of aldehydes and ketones should be provided.

Reviewer: 3

Comments to the Author(s)

The revised version seems fine.

Author's Response to Decision Letter for (RSOS-210142.R0)

See Appendix B.

Decision letter (RSOS-210142.R1)

Dear Dr Bracca:

Title: Eco-friendly methoximation of aromatic aldehydes and ketones using $\text{MnCl}_2 \cdot 4\text{H}_2\text{O}$ as an easily accessible and efficient catalyst

Manuscript ID: RSOS-210142.R1

It is a pleasure to accept your manuscript in its current form for publication in Royal Society Open Science. The chemistry content of Royal Society Open Science is published in collaboration with the Royal Society of Chemistry.

Yours sincerely,

Dr Laura Smith

Publishing Editor, Journals

On behalf of the Subject Editor Professor Anthony Stace and the Associate Editor Dr Annette Trunschke.

RSC Associate Editor
Comments to the Author:
(There are no comments.)

Reviewer(s)' Comments to Author:

Appendix A

Dra. Andrea B. J. Bracca,
INSTITUTO DE QUIMICA ROSARIO
CONICET - Universidad Nacional de Rosario
Suipacha 531 - S2002LRK Rosario - Argentina
Tel.: 54 - 341 - 437-0477, int. 103
E-mail: bracca@iquir-conicet.gov.ar

Rosario, January 27, 2021

Dear Editor,

We have received the message containing your decision with regards to our manuscript (RSOS-201654). As you mentioned in the mail, there are two conflicting opinions, but you have indicated that your decision was based on the lack of a performance comparison, asked by both Reviewers and the opinion of Reviewer 1 (lack of novelty), whose exact words were: “any exciting aspects are *not well included*”, “without conceptual large innovative steps” and “it is not very general and deep comparisons [...???”).

We also thank your answer to our mails, sent in order to clarify some points before submitting our revision. Through your today’s message (“I can confirm that unfortunately there was indeed an administrative error whereby the Supporting Information file was not supplied to the reviewers”), we have confirmed the source of a big part of the reviewers’ requests.

Regarding the Supporting Information file, I would like to offer a plausible explanation and my apologies, if needed. The originally submitted manuscript required some language and text polishing to pass the technical check. We conjecture that at the time the polished version was uploaded, the Supporting Information file accompanying the original submission was somehow deleted and not reuploaded, so it was not available for review. Hence the requests made by both reviewers.

In order to provide a proper answer to both reviewers, this revised version is uploaded along with the corresponding Supporting Information file, which includes the couple of detailed Tables (see pages S30-S32) found in its original version. The Tables contain the requested comparison for the same compounds we synthesized, at the state of the art. In one of the Tables a comparison with the general literature is made, whereas in the other, we compared the performance of the Mn(II)-based method with that of our previously published Ce(III)-based approach.

The first Table clearly reflects the importance of our findings and the benefits of our method compared with the general literature (reduced amounts of reagents, lower temperature, less time, better yields, avoidance of objectionable reagents/solvents, etc., and even an easier work-up). Perhaps our method does not get all the records (reduced..., lower..., less..., better...) simultaneously and in all cases, but a quick look at the Table and comparison with the results in the paper (or in the second Table) should clearly inform the reader that our method makes a difference. In addition, drawbacks of existing methods and the need of alternative and improved methodologies are now more clearly and extensively stated in the manuscript.

In addition, in the comparison with the state of the art it should be also taken into account that the method has several positive distinctive features: 1- It is eco-friendly by design, since $\text{MnCl}_2 \cdot 4\text{H}_2\text{O}$ and NaOAc are classified as GRAS chemicals [compare for instance with a single titanium based paper, or even with our own Ce(III) method]; 2- It employs the less toxic and inexpensive ethanol as solvent (MeOH and 2-PrOH are more expensive and more toxic); 3- It is comparatively very efficient (in general, better than available alternatives, or a suitable complement for them); 4- It uses an economic and easily accessible catalyst, at a low catalyst load and without the requirement of a special external ligand (these aspects were correctly captured by Reviewer 2), also leaving some room for improvement (out of scope of the current manuscript). Therefore, we understand that these Tables, two key pieces of information, should fully satisfy both reviewers.

Regarding the novelty issue raised by the first reviewer, at least the following aspects mentioned in the manuscript should be taken into account:

1- Some more expensive aniline derivatives have been developed for use in the area of biorthogonal chemistry. However, they have specific features and requirements (compatible with cells life, dilute solutions, useful at essentially neutral pH, etc.). Besides them, there are essentially no studies regarding the catalysis/acceleration of the methoximation reaction, as confirmed by an additional Reaxys search. Our work is novel because it comes to fulfil an important objective need for catalytic methoximation options.

Dra. Andrea B. J. Bracca,
INSTITUTO DE QUIMICA ROSARIO
CONICET - Universidad Nacional de Rosario
Suipacha 531 - S2002LRK Rosario - Argentina
Tel.: 54 - 341 - 437-0477, int. 103
E-mail: bracca@iquir-conicet.gov.ar

CONICET

I Q U I R

2- The metal-mediated oximation of carbonyl derivatives has been scarcely reported. Further, as stated in the Introduction, in a recent review it was observed that such catalysis of the reaction was “still-uncommon” and that “the synthetic potential of many of these approaches is not high”. The first part of the statement is correct, turning our research into an original piece. Further, our results are also novel and will be useful, because they clearly disprove the last assertion.

3- Our findings include the novelty that a GRAS chemical such as $\text{MnCl}_2 \cdot 4\text{H}_2\text{O}$ can catalyze the oximation of aromatic (and aliphatic) aldehydes and ketones (in the presence of a GRAS base such as NaOAc, instead of pyridine), in the less toxic solvent ethanol. Besides being novel, they also are of interest because the process has a wide scope, as correctly interpreted by Reviewer 2.

4- Despite $\text{MnCl}_2 \cdot 4\text{H}_2\text{O}$ has found some limited use in organometallic chemistry (mainly for C-C forming reactions), this is the first time the salt is proposed as an efficient methoximation catalyst at a low catalyst load and without added ligands, in a novel and minimalistic reaction design.

We favor simplicity, sustainability, economy, eco-consciousness and effectiveness. We do not believe that a catalyst must be based on an endangered metal or have complex and expensive ligands for being useful or novel. The state of the art may also be advanced by novel and useful applications of known compounds.

The above details are included in the manuscript, as our most powerful justification of its originality and novelty. It is also hoped that our findings will act as a starting point for fine-tuning the basic catalyst (i.e., finding proper ligands to improve kinetics and reduce catalyst load).

Looking forward to your soonest and positive reply, I remain,

Sincerely,

Dr. Andrea B. J. Bracca

Appendix B

Rosario, May 12, 2021

Dear Editor,

Thank you for your May 12 message, informing that our manuscript ID RSOS-210142 has been accepted for publication, pending minor revision, consisting in three observations made by Reviewer #1 (Reviewer #3 considered the manuscript as suitable for publication in its current form). The following are the reviewer's questions and our corresponding answers:

Question # 1: Why $\text{MnCl}_2 \cdot 4\text{H}_2\text{O}$ provides the best catalytic activity for methoximation of ketones compared to other manganese salts?

Our answer #1: We have already answered this question in the manuscript. In paragraph 3 of the Results and Discussion section, we inform that "Although it may be argued that addition of NaOAc to the reaction may afford $\text{Mn}(\text{OAc})_2$, it was found that the counterion seems to be relevant, since $\text{MnCl}_2 \cdot 4\text{H}_2\text{O}$ turned the transformation most efficient, resulting in full depletion of the starting ketone in only 30 min". Further, when discussing the mechanism, the following text can be read: "Plausibly, the extent of the coordination depends on the nature of the counteranion in the Mn(II) Lewis acid catalyst, which would explain the differences in reactivity between MnSO_4 and MnCl_2 ".

Question #2: How about the purity of the manganese chloride salt? Does the purity influence the catalytic activity?

Our answer #2: We have used analytical grade reagents acquired to Aldrich Chemical Co. (now part of Merck). We have made a more detailed statement about reagent quality in the text of this revised version. As with others, in the specific case of MnCl_2 , a Reagent Plus grade was used (> 99%). On the other hand, in our previous paper (properly referenced in the manuscript), we have made an ample catalyst screen and found out that $\text{CeCl}_3 \cdot 7\text{H}_2\text{O}$ and $\text{MnCl}_2 \cdot 4\text{H}_2\text{O}$ were the most promising promoters. MnCl_2 is a rather inexpensive reagent, which is currently available in high purity; therefore, the possible effect of the most common metal ions should be discarded. Hence, we understand that there is no point to study the effect of impurities on the catalytic activity (they should be at least 100 times more efficient than our proposed catalyst), unless the nature of the impurity is previously known. Under the experimental conditions reported in the manuscript, our work should be fully reproducible anywhere.

Question #3: Comparison with other catalysts reported in the literature for methoximation of aldehydes and ketones should be provided.

Our answer #3: The answers required by the reviewer can be found in the manuscript and the Supplementary Information file, where they have been from the initial submission (and more detailed information was added in our previous revision). In the manuscript, before the insertion point suggested for Figure 1 (upper part of page 2) we discuss about aniline derivatives employed in biorthogonal chemistry (different context and conditions) and in the next three paragraphs we provide information about Ti, Fe and Ce derivatives with the corresponding references. In addition, the long Tables S1 and S2 at the end of the Supplementary Information file contain the requested performance comparisons, along with the corresponding references.

We think that we have answered all of the reviewers observations and that have provided satisfactory answers, and that now, our accepted manuscript should be considered as suitable for publication.

Looking forward to your soonest and positive reply, I remain,

Sincerely,

Dr. Andrea Bracca